# Defense-Oriented Psychoanalytic Psychotherapy as a Tailored Treatment for Boys: Neurobiological Underpinnings to Male-Specific Response Tested in Regulation-Focused Psychotherapy for Children

**DOI:** 10.3390/bs12080248

**Published:** 2022-07-22

**Authors:** Timothy Rice, Tracy A. Prout, Andreas Walther, Leon Hoffman

**Affiliations:** 1Icahn School of Medicine at Mount Sinai, New York, NY 10029, USA; hoffman.leon@gmail.com; 2Ferkauf Graduate School of Psychology, Yeshiva University, Bronx, NY 10461, USA; tracyprout@gmail.com; 3Department of Clinical Psychology and Psychotherapy, University of Zurich, 8050 Zurich, Switzerland; a.walther@psychologie.uzh.ch; 4New York Psychoanalytic Society & Institute–Pacella Research Center, New York, NY 10028, USA

**Keywords:** men’s mental health, boys, emotion regulation, neurodevelopment, oppositional defiant disorder, psychoanalytic psychotherapy, defense analysis

## Abstract

This paper presents defense-oriented psychoanalytic psychotherapy as a tailored treatment for boys through a neurophysiological hypothesis. Male central nervous system development is reviewed, with a focus on the development of the emotion regulation system. The organizational effects of pre- and post-natal androgens delay central nervous system development in males relative to females, following a caudal to rostral phylogenetic framework. Ventromedial prefrontal structures mature at an earlier developmental age than dorsolateral prefrontal structures, creating less of a gender gap in the available underlying neural architecture for responsivity to targeted therapeutic intervention. The hypothesized operation of defense analysis upon ventromedial prefrontal cortical structures and corticolimbic connectivity therefore positions boys to benefit from psychotherapy equally as girls. In this study, we explored gender differences in presentation and response to a short-term, manualized defense-oriented psychoanalytic psychotherapy named regulation-focused psychotherapy for children. In a sample size of 43 school-aged children, consisting of 32 boys and 11 girls, with oppositional defiant disorder, we found no statistically significant differences in participant characteristics upon entry nor in treatment response, as measured by changes in scores on the Oppositional Defiant Disorder Rating Scale, the oppositional defiant problems subscale of the Child Behavior Checklist, the suppression and reappraisal subscales of the Emotion Regulation Questionnaire for Children and Adolescents, and the lability and negativity subscale of the Emotion Regulation Checklist. The findings were comparable with the gendered findings of preexisting studies of play therapy, where boys and girls improve equally, but not of behaviorally predominant psychotherapy, where girls appear to have superior responses. Our findings suggest that the treatment as a general play therapy, but with a focus on the implicit emotion regulation system, was successful in meeting boys’ gendered treatment needs. Conclusions are drawn with implications for further study.

## 1. Introduction

A men’s mental health perspective among children and adolescents [1,2] focuses upon male-specific considerations in formulation and care delivery. In matters of psychotherapy, this includes gendered understandings of symptomatology present upon initiation of care, intervention planning, and therapeutic responses. To identify a treatment as tailored for boys, it must be relevant to the problems they face, work in their language, and create tangible benefits. In this article, we position defense-oriented psychoanalytic psychotherapy as an attuned treatment for boys on account of its hypothesized mechanism of action.

Specifically, defense-oriented psychoanalytic psychotherapy is applicable to a broad range of psychopathologies through its operation upon the implicit emotion regulation system. The implicit emotion regulation system in boys appears to be an ideal target of intervention relative to other neural systems.

This paper reviews the underpinnings of this hypothesis and tests these foundational assumptions through data obtained in a randomized controlled trial of one defense-oriented psychoanalytic psychotherapy, termed regulation-focused psychotherapy for children (RFP-C).

### 1.1. Developmental Hypothesis

Boys relative to girls experience temporally slower central nervous system maturation [3]. This occurs as a function of the organizational effects of pre- and post-natal testosterone and yields neurophysiologic and behavioral consequences, including for childhood emotion regulation and aggression [4].

Emotion regulation is a prefrontal cortico-limbic-dependent executive function defined as the capability to adjust emotions to environmental demands [5]. Broadly, emotion regulation occurs through higher-order prefrontal modulation of lower-order limbic and brainstem structures [6]. Deficits in emotion regulation play a role in a wide range of psychopathology [7], and promoting adaptive emotion regulation development may be understood as a transdiagnostic approach to address childhood pathology. As child psychotherapy using the modality of play can positively impact a wide range of executive functions [8], so may psychoanalytic psychotherapy targeting this executive function advance healthy development.

Explicit emotion regulation, in which emotions are modulated consciously and effortfully, and implicit emotion branch, in which emotions are modulated unconsciously and automatically, are two separate processes which comprise emotion regulation [9]. These two methods of emotion regulation originate from different areas of the prefrontal cortex and show different underlying neural paths to modulated limbic areas [10]. Prefrontal explicit emotion regulation correlates originate dorsolaterally, whereas implicit emotion regulation correlates originate more ventromedially [10].

Cognitive behavioral psychotherapeutic strategies targeting the explicit emotion regulation branch have already been demonstrated to alter the underlying neurophysiology of emotion regulation [11]. These interventions employ didactic-based instructions in anger management training, by operating upon and resulting in changes in the functioning of the dorsolateral prefrontal cortex [11]. However, this didactic style of training intervention may be less acceptable to boys than to girls relative to more flexible, casual approaches [12]. Males are commonly understood to be action-oriented, less verbally and socially attuned, and perform the work of psychotherapy through the experiential medium [12]. Additionally, developmentally, children are not yet positioned to make full use of cognitive behavioral interventions associated with interventions upon training higher adaptive explicit emotion regulation strategies [13].

These concerns notwithstanding, there is a need to help boys with emotion regulation: School-aged boys have lower emotion regulation capabilities relative to school-aged girls [14]. Neuroimaging studies demonstrate that the development of the prefrontal cortex, from where emotion regulation arises, is delayed in boys relative to girls [15]. In girls, the development of neuromodulation processes reach maturity approximately two years earlier than in boys [3]. There is a pressing need to develop psychotherapeutic interventions acceptable to boys that would help promote the development of emotion regulation towards its adaptive functioning.

Central nervous system development proceeds prenatally to the third decade of life [16]. It proceeds from synaptogenesis from prenatal through the toddler years, to pruning to late adolescence, and to myelination to early adulthood [16]. The temporal distribution of its developmental stages has relevance to the distinction between explicit and implicit emotion regulation. Maturational processes occur phylogenetically first in the ventromedial prefrontal cortex relatively to more dorsolateral areas [16]. Children therefore attain the underlying neural correlates sufficient for mature, developed implicit emotion regulation prior to those of explicit emotion regulation, given that the ventromedial regions of the prefrontal cortex from where implicit emotion regulation arise develop prior to the dorsolateral regions of the prefrontal cortex [16].

Young men’s lack of developed dorsolateral prefrontal areas relative to their age-matched female peers may be the underlying neural correlate of their lowered expectations for [17], and less progress within [18], traditional psychotherapy, which can include cognitive behavioral interventions. Their relative attainment of ventromedial maturity relative to their age-matched female peers may position them to benefit more equably from interventions targeting this underlying neural architecture and physiology. Just as cognitive behavioral psychotherapies may target self-regulation and its underlying prefrontal substrates [11], so too may targeted interventions from the psychoanalytic canon, which focuses on unconscious, automatic, and effortlessly initiated adaptive and maladaptive processes as a core tenet of health, pathology, and human subjectivity.

There are many implications of this gender gap in both presentation and treatment. For example, it could be proposed that the male predominance in prevalence within the common childhood disorder of attention deficit/hyperactivity disorder [19], which is so dependent upon prefrontal correlates [20], could be related. The Western cultural expectations upon girls to be more contained, and in greater control relative to boys, may also be an important factor. In regard to treatment, mother’s permissiveness of aggression in boys relative to girls [21], which may subsequently delay seeking treatment, is another potential manifestation.

We propose that targeting of the implicit emotion regulation system in psychotherapy interventions may lessen the developmental gap between the genders in the ventromedial prefrontal regions relative to dorsolateral regions. This would make such psychotherapy more appropriate for boys, by the hypothesis that the underlying neurophysiological correlates available for traditional cognitive behavioral interventions are not sufficiently developed in male children, relative to age-matched female children, to be adequately receptive to intervention: boys may not yet be neuroanatomically equipped to receive the lessons of didactic-style skills training by virtue of their underdeveloped dorsolateral prefrontal cortical areas relative to girls. Because the ventromedial areas that are the correlates of implicit emotion regulation are chronologically relatively more developed, boys would have to a greater extent the same underlying neurophysiological correlates available for intervention as girls, making defense-oriented psychoanalytic psychotherapy, which targets these areas, more appropriate. Later, when boys’ central nervous systems’ development “catches up” to that of girls, as occurs following puberty [15], a greater spread of opportunities for psychotherapy exist. Even in adulthood, men may rely on automatic emotion regulation to a greater extent than women [22], suggesting a defense-oriented psychoanalytic psychotherapy for boys to be distinctly well positioned for formulation as an intervention responsive to the male worldview.

Organizations such as the American Psychological Association have identified the need to tailor psychotherapeutic approaches to males through the creation of guidelines for psychological practice with boys and men [23]. This paper responds to this need by situating defense-oriented psychoanalytic psychotherapy as such a treatment intervention. To explore this proposal, we examine the prior meta-analyses of efficacy studies of psychotherapy with children and adolescents, including those using the medium of play.

### 1.2. Prior Efficacy Meta-Analyses

The first meta-analysis of child therapy outcome studies was conducted in 1985 [24], and identified 64 studies of psychotherapy with children under 12. The majority (37) were behavioral, and the minority (5) were psychodynamic. Studies with a majority of male participants were found to have smaller effect sizes. The next meta-analysis to evaluate gender effects specifically in children occurred in 1987 [25], and again found decreased effect sizes for boy majority groups (average *d* = 0.80) relative to girl majority groups (average *d* = 1.11; *p* = 0.33). Though the authors of this early meta-analysis did not report on statistical significance concerning the large discrepancy in average effect sizes, the findings suggest that boys have reduced responses to psychotherapy relative to girls. In this study there was again predominance of behavioral studies included in the meta-analysis: 197 were behavioral, while 27 were non-behavioral, including 9 which were insight-oriented.

Whereas the next meta-analysis of child therapy, which occurred in 1990, did not evaluate outcomes by gender [26], the following study in 1995 [18] which included adolescents found that, though female adolescents responded better again to behaviorally predominant psychotherapies than male adolescents (*p* < 0.0001), no differential treatment response by gender was found for children. A total of 150 studies with 244 different treatment groups had been included, of which the majority (197) were classified as behavioral and consisted of operant, respondent, modeling, social skills, cognitive/cognitive behavioral therapy (CBT), parent training, multiple behavioral, and other behavioral approaches. Again, only 9 were insight-oriented, while 36 of the 197 behavioral therapies were parent training.

These early meta-analyses suggest that behavioral interventions for children tend to produce strong results among female children, but less so for male children, when the interventions are predominantly behavioral. In the last study, that parent training was included in greater number relative to the earlier studies suggests that the novel finding of equivalency between boys and girls may be a product of working through the parent. The authors of the 1995 study [18] noted that the favorable therapeutic response among adolescent females, as compared to males, may reflect female adolescents’ heightened interpersonal skills, that allow them to make better use of the relationship with the therapist. Additionally, the authors noted that increasing trends favoring girls’ responsiveness to treatment after the mid-1980s may imply that treatments have become more sensitive to the characteristics and treatment needs of girls.

Subsequent meta-analyses from 2001 [27], 2005 [28], and 2015 [29] specifically examined play therapy as a treatment modality. They found no differences in response across gender for play therapy with children. When the medium of play as drawn from the psychoanalytic and client-centered traditions was specifically explored, rather than CBT modalities, no gender differences were found.

These findings demonstrate the importance of designing psychotherapies adapted to meet the unique needs of male children. The avenue of play is a unique medium operating outside of the explicit emotion regulation strategies emphasized in CBT interventions. When play interventions are operationalized to a specific strategy to target implicit emotion regulation, theoretical propositions suggest great benefits to boys are possible. The RFP-C therapeutic approach is conceptualized as such.

### 1.3. A Defense-Oriented Psychoanalytic Psychotherapeutic Approach

Regulation-focused psychotherapy for children is a short-term, manualized psychoanalytic psychotherapy for school-aged children with externalizing disorders that focuses on the interpretation of children’s observable defense mechanisms against painful affects [30]. The approach was formulated to be in alignment with the understanding of the male-specific neurobiology of boys [31], by noting a parallel between defense mechanisms and implicit emotion regulation [32].

Specifically, the approach understands the oppositionality and defiance of externalizing disorders to be the behavioral manifestations of developmental lags in the implicit emotion regulation system. Throughout the treatment, the proposed active intervention consists of systematic, iterative interpretation of children’s defenses, thereby differentiating it from traditional client-centered play therapy and cognitive behavioral approaches [33]. As a clinical example, in work with a boy who would repeatedly cheat in checkers at moments when the overlaying conversation aroused uncomfortable feelings, the therapist would serially comment that the cheating (observable, “disruptive” behavior) served to protect him (defense) from discomfort (affect) surrounding what was being discussed.

In another example [34], when a school-aged foster child playing ball with a clinician begins to throw the ball with more force at the clinician after the clinician mentions that the session will soon end, the clinician comments that he notices the wild throwing of the ball (defense) began after s/he mentioned the session would end, asking aloud what the child made of that? With time, the clinician can comment that the ending of the sessions and loss of the provider brings painful feelings, and the child can be helped to understand that his actions serve to protect himself against from these feelings (passive into active, identification with the aggressor). Comments upon drive conflicts and fantasies do not occur. In this vignette, one assessment may be that the forceful throwing of the ball represents an enacted projective process in relation to a conflict between an aggressive wish to hurt the clinician and a wish to jettison aggressive urges towards the ambivalently held clinician. For example, these understandings can inform conceptualizations of the child’s understanding of loss and departures as aggressive and as patterned upon his history in the foster care system. This can allow the clinician to be sensitive to derivatives of this material in the treatment and be attuned to these areas for defenses to interpret in an experience-near manner.

Differentiation and specification of affect with increasing understanding of thematic content particular to the child as it unfolds within the treatment follows. Focusing the treatment on this mode of intervention within the security of a strong therapeutic alliance promotes alternative and more adaptive processes of implicit emotion regulation. This proceeds by attenuating distress associated with avoided affects through implicit normalization (offering that these feelings are normal to experience), validation (offering that it is subjectively valid to feel these feelings), and universalization (offering that many children feel these feelings), and through automatic processes of internalization models in the therapeutic matrix for healthy emotional regulation.

This model is in accordance with the Research Domain Criteria model [35], which links the active intervention to a hypothesized specific action upon underlying neural correlates with observable behaviors. Symptoms were significantly reduced in children with opposition defiant disorder (ODD) during a randomized controlled trial [36] of this therapeutic approach. Oppositional defiant disorder is among the most common reasons for childhood referral to specialty clinics [37], and ODD is characteristically a common problem of young men. The boy to girl ODD ratio globally is consistently about 2:1 (British Child and Adolescent Mental Health Survey 4.8%: 2.1% [38], The Great Smoky Mountains Study 3.1%:2.1% [39], The Bergen Child Study 2.0%: 0.9% [40]). Not only is ODD more prevalent in boys than in girls, but professional help is also more commonly sought for boys than girls with ODD [41]. Of the 43 children aged 5 to 12 in the study with ODD, 32 were boys, perhaps representative of parents seeking treatment for ODD for boys more commonly than for girls.

The initial randomized controlled trial of RFP-C found significant reductions in our primary outcome measure of ODD symptoms as measured by the Oppositional Defiant Disorder Rating Scale (ODD-RS [42]) (*p* < 0.001, *d* = 1.4) [36]. When comparing active treatment to waitlist, no significant changes in ODD symptoms, as measured by the oppositional defiant problems subscale of the Child Behavior Checklist (CBCL) [43] or in measures of emotion regulation via the Emotion Regulation Questionnaire for Children and Adolescents (ERQ-CA) [44], and the Emotion Regulation Checklist (ERC) [45], were identified. Significant improvements on the oppositional defiant problems subscale of the CBCL were observed when examining all treatment completers as a whole, including those who participated in the waitlist condition prior to initiating treatment (*p* < 0.001, *d* = 0.95).

To date, no evaluation of the gendered characteristics of the participants or their outcomes has been employed. As we have proposed defense-oriented psychoanalytic psychotherapy as a tailored treatment for boys based on the neurophysiologically informed theoretical propositions above, we aimed to examine whether boys and girls would benefit equally from RFP-C. Specifically, we chose to explore:Would boys respond equally to girls, as measured by our primary outcome measure, pre- versus post-treatment scores on the Oppositional Defiant Disorder Rating Scale?Would boys respond equally to girls, as measured by a secondary outcome measure focusing on ODD symptomatology, pre- versus post-treatment scores on the oppositional defiant problems subscale of the Child Behavior Checklist?Would boys present with similar or disparate ER (as compared to girls) at pre-treatment and post-treatment, as measured by scores on the lability and negativity and overall emotion regulation subscales of the Emotion Regulation Checklist?Would boys present with similar or disparate ER (as compared to girls) at pre-treatment and post-treatment, as measured by the suppression and reappraisal subscales of the Emotion Regulation Questionnaire for Children?

Based on the results of the prior meta-analyses of play therapy [27,28,29], where no differences in response across gender were found, we anticipated to find no statistically significant differences between the groups of boys and girls of our trial in ODD symptomatology by our primary and secondary outcome measures (hypotheses 1 and 2). Because we believe ER relates to ODD symptomatology, we anticipated to find no significant differences between the groups of boys and girls in ER measures, both in relation to measures of overall ER as well as of explicit ER. The final hypothesis was exploratory in that our active intervention upon the implicit emotion regulation system is not hypothesized to directly change measures of explicit emotion regulation, although downstream effects or associations may be observed, In the discussion, we comment on gendered elements of our trial and offer directions for future study.

## 2. Materials and Methods

Full details of the study are detailed in the original study manuscript [36] and summarized in an abridged version and with greater detail concerning gender considerations below.

### 2.1. Participants

Participants in this study were 43 school-aged children (mean age = 7.84, ±1.95; range 5–12 years, 32 boys and 11 girls, none were gender expansive) with a primary diagnosis of ODD as confirmed in baseline assessments by the Children’s Interview for Psychiatric Syndromes–Parent Version (P-ChIPS) [46], a score of ≥8 on the ODD-RS [42], and clinically significant distress on the oppositional defiant problems subscale of the CBCL [43].

### 2.2. Therapists

Therapists were 22 doctoral candidates, of whom 21 were female. The therapists were a convenience sample of trainees in the laboratory group of the study authors. Two were in their first year of training, five were in their second year of training, seven were in their third year of training, and eight were in their fourth year of training. Additional details on the cohort are described in the original study manuscript [36]. Therapists were trained in the protocol in small group format, and supervision was provided by the study authors (T.R., T.A.P., L.H.).

### 2.3. Procedures

Participants were recruited from urban and nearby suburban communities for a baseline assessment including the P-ChIPS, the ODDRS, the CBCL, the ERQ-CA [44], and the ERC [45]. Those who met inclusion and exclusion criteria (see original study manuscript) were randomized to a 10-week waitlist control or to active intervention.

The waitlist participants discontinued alternative psychosocial interventions and those administered psychiatric medications agreed to make no substantial medication changes during the waitlist or active treatment conditions. At the end of the 10-week waitlist, they re-completed the assessments administered at baseline and began the active intervention. A total of 22 subjects were randomized to the waitlist group, of whom 20 completed the waitlist. The average age was 8.09 years, and 18 of the 22 subjects were boys (81%). Following a cross-over design, all waitlist completers were invited to subsequently enroll in the active intervention.

The active intervention consisted of the manualized, twice-a-week psychotherapy course of RFP-C, consisting of a total of 16 child sessions and 4 parent sessions. At the end of the active intervention, the participants again completed the assessments administered at baseline. In total, 21 subjects were randomized to the active intervention group, of whom 18 completed the treatment. The average age was 7.57 years, and 14 of the 21 subjects were boys (67%).

Of the 18 children who began treatment after waitlist, 16 completed treatment. This led to a total of 34 treatment completers, of which 18 were from the active group, and 16 were from the waitlist with delayed treatment group.

### 2.4. Data Analysis

The current study explored gender differences pre- and post-treatment across several outcome variables. Descriptive statistics including group average scores for several variables were performed.

To evaluate measures of emotion regulation, pre- and post-treatment averages for suppression and reappraisal were drawn from the ERQ-CA from the boys’ and the girls’ groups, while measures of lability and negativity and overall ER were drawn from the ERC. Two-tailed independent samples *t*-tests were used to examine potential differences between group averages. Independent samples *t*-tests were also used to examine differences in pre- and post-treatment symptom scores for boys and girls. One-way ANOVAs were used to evaluate ER measures. To evaluate group differences in measures of impairment and the presence of psychiatric medications, chi-squared analyses were performed.

## 3. Results

### 3.1. Pre-Treatment Differences in Demographics and ODD Symptomatology by Gender

Prior to testing for differences in treatment outcome and differences in emotion regulation at post-treatment, we examined several key variables at baseline as presented in Table 1. There were no statistically significant differences for age between boys (*M* = 7.72, *SD* = 1.91) and girls (*M* = 8.18, *SD* = 2.14) in this study (*t*(41) = 0.68, *p* = 0.56). The presence of attention deficit/hyperactivity disorder (ADHD) was assessed at the start of treatment, given that it is a common co-occurring disorder in ODD and may potentially impact ER scores. There were no differences in prevalence of ADHD at baseline, as measured by the CBCL, with 63% of girls and 64% of boys presenting with clinically significant symptoms of ADHD at intake (*t*(41) = 0.60, *p* = 0.32). Similarly, there was no statistically significant difference between the boys and girls in overall impairment at intake, as measured by the CBCL (*t*(41) = −1.32, *p* = 0.20) and the ODDRS (*t*(41) = 0.52, *p* = 0.75). Finally, there was no statistically significant difference in rates of prescription of psychiatric medication between boys and girls, (χ^2^ (1, N = 42) = 2.98, *p* = 0.08). Of note, although the chi-square test for prevalence of psychiatric medication use was not statistically significant, of the seven children who were taking medication during the study, all were male.

### 3.2. Treatment Outcomes by Gender

There was no statistically significant difference in changes on the primary outcome measures (ODDRS and CBCL), which assessed symptoms of ODD at baseline and post-treatment. Boys’ scores on the ODDRS declined by an average of 7.65 points and girls had an average improvement of 5.91 points (range = 0–24, *t*(32) = 0.91, *p* = 0.39). Similarly, there were no differences by gender for improvement on the ODD subscale of the CBCL (*t*(26) = −0.04, *p* = 0.97). The reduction in degrees of freedom between the ODDRS and the ODD subscale of the CBCL was a product of reduced response rates with the CBCL, as it was a more time-demanding form to complete than the eight-item ODDRS, and completion required returning the form by physical mail during the COVID-19 pandemic.

### 3.3. Emotion Regulation

One-way ANOVAs were used to examine differences in ER between boys and girls at pre-treatment and post-treatment; there were no statistically significant differences between the groups in pre or post-treatment measures of emotion regulation, as assessed by the ERC and the ERQ-CA. Specifically, there were no significant differences on parent-reported, pre-treatment lability and negativity (*F*(1, 41) = 3.04, *p* = 0.09) or overall emotion regulation (*F*(1, 41) = 0.09, *p* = 0.77) for boys versus girls on the ERC. Similarly, child/adolescent-reported pre-treatment ER was comparable across genders for suppression (*F*(1, 40) = 0.01, *p* = 0.92) and reappraisal (*F*(1, 39) = 0.51, *p* = 0.48) on the ERQ-CA. Similarly equivalent results were found across genders at post-treatment for lability/negativity (*F*(1, 28) = 0.18, *p* = 0.67), overall emotion regulation (*F*(1, 28) = 1.40, *p* = 0.25), suppression (*F*(1, 27) = 2.07, *p* = 0.16), and reappraisal (*F*(1, 27) = 0.19, *p* = 0.67). Graphic representation of the change in all four ER variables by gender is presented in Figure 1, Figure 2, Figure 3 and Figure 4 below.

## 4. Discussion

This study evaluated gender differences among school-aged children with ODD in a trial of a defense-oriented psychoanalytic psychotherapy for school-aged children, RPF-C. Results of the current study suggest that there were no significant differences between boys and girls who participated in a recent trial of RFP-C. Based in the medium of play, RFP-C would be expected to be as efficacious for boys as for girls. Our finding of no statistical significance in treatment response between boys’ and girls’ groups is concordant with the meta-analyses of play therapy in children [27,28,29]. As such, our first three hypotheses were supported, whereas our investigation of our fourth exploratory hypothesis yielded data for future investigation.

### 4.1. Pre-Treatment Differences by Gender

There were no significant pre-treatment differences by gender. Descriptively, the slightly advanced age of the girls relative to the boys is supported by the perspective that professional help may be commonly sought for boys earlier than for girls, as was found in one study [47]; it could be hypothesized that the girls were referred at a later age owing to tendencies in culture and presentation that lead boys to seek care for ODD to a greater extent. It was reassuring to find no difference in averages of concurrent ADHD (63% versus 64%), suggesting that the presence of this highly comorbid disorder did not account for differences in the results.

It is notable that a nearly significant finding (*p* = 0.08) existed concerning medication status, favoring the medication of boys relative to girls; of the seven children who were prescribed medication who were enrolled into the study, all were male. The presence or absence of disorders which are more commonly treated with pharmacotherapy, such as ADHD, cannot account for this finding. A hypothesis regarding this finding that warrants future study concerns the role of the parents in electing to proceed with medication. Parents may tolerate less well the predominantly physical aggression of boys with ODD relative to the predominantly more subtle and relational aggression of girls with ODD, and thus proceed with medication to seek help for these symptoms to a greater extent [48]. Analysis of qualitative data contained in the transcripts of the children may identify supportive themes and may be a topic for further study. Determination of the type of medication prescribed, the process of how it was selected, and other factors between the parents in proceeding with the choice may yield important data.

### 4.2. Therapeutic Response

No statistically significant difference emerged between the boys and the girls in their response to the treatment on the primary outcome measure of ODD-RS score change. While there was, descriptively, a greater improvement among the boys (average change 7.65) relative to the girls (average change 5.90), only further study with a larger sample size could evaluate for any meaning to this finding. At present, our results suggest that boys respond to treatment similarly to girls, which is consistent with the findings of meta-analyses of play psychotherapy [27,28,29].

Limitations include that we did not evaluate for covariates in our analyses. The descriptive difference in age would be a worthwhile covariate to explore in light of our developmental hypothesis: because we believe age-matched boys to have a lag relative to girls in the development of underlying neuronal architecture for emotion regulation capabilities that are responsive to therapeutic intervention, the discrepancy between our groups by age would further widen this gap. Medication status on account of its nearly significant difference would be another worthwhile covariate.

### 4.3. Emotion Regulation

No differences in emotion regulation between the groups pre-treatment were found. This finding is notable in that it conflicts with the findings that school-aged boys have lower emotion regulation capabilities relative to school-aged girls [14]. This notwithstanding, nearly significant differences in the negativity and lability scores pre-treatment (*p* = 0.09) held to expectations that boys are generally more negative and labile as products of emotion regulation deficits relative to girls. The non-significant post-treatment differences in these scores (*p* = 0.18) may benefit from further future study with a larger sample to further evaluate for significance, though the pre-treatment differences may be a confounder.

Although both suppression and reappraisal as ER strategies are adaptive when employed flexibly in appropriate circumstances [49], an overreliance upon suppression as an explicit emotion regulation strategy is generally understood to be less adaptive than an overreliance upon reappraisal. The two strategies differ in their neural signatures temporally in the prefrontal resources required for deployment and in their effectiveness in reducing limbic arousal [50]. The finding that reappraisal scores descriptively decreased for both boys and girls after treatment while symptomatology descriptively reduced suggests that the treatment, focusing on implicit emotion regulation, may have helped the children to use alternative strategies of implicit emotion regulation and require less of a need to employ explicit strategies.

The descriptive decrease in suppression scores for boys, but descriptive increase in suppression scores for girls, may suggest that boys were more able to make use of the implicit emotion regulation interventions than girls. A significant limitation of our ability to draw conclusions from this data is the lack of a measure within our data set of a measure of implicit emotion regulation skills, such as may be provided by a coding of a hierarchy of defense mechanisms [51]. This could be performed in the first and last sessions for a pre- and post-treatment analysis, as well as through a continuous application through the course of a treatment for the children to evaluate for longitudinal changes. Further study of the dataset may help to explore this question.

## 5. Conclusions

Efforts in the literature to emphasize the importance of engaging men in psychotherapy [52] identify the importance of specific process skills for engaging men. Mahalik and colleagues [53] surveyed 475 members of the American Psychological Association to develop a taxonomy of helpful and harmful practices for clinical work with boys and men. Others [54,55] have noted the importance of such work in psychotherapy with boys and young men. These efforts become increasingly important owing to the increasingly common patient–provider gender discrepancy when working with boys and men, as psychology as a field becomes increasingly a field of female providers [56]. The implications of these demographic trends have been highlighted since at least 1992 [57], and underscore the importance of tending to the gendered needs of boys and young men.

This study shows that the systematic, experience-near interpretation of defenses against painful affects that comprises the core of RFP-C can sensitively tend to young men. We have shown that this core intervention is hypothesized to operate on areas of the brain which may be more available to psychotherapeutic work than those that are targeted by cognitive-behavioral interventions. Our findings show that boys experience no statistically significant difference in responsiveness to this intervention relative to girls. Our figures demonstrate some non-statistically significant findings suggestive that boys, relative to girls, may in fact respond favorably in relation to lability and negativity and overall emotion regulation. Future studies expanding our sample sizes may revisit these questions and reassess for statistical significance. At this time, we can recommend clinicians provide psychotherapy incorporating play for boys which can integrate and experiment with our defense-oriented approach.

Regulation-focused psychotherapy for children is one of many approaches in the body of comprehensive rehabilitation approaches incorporating psychotherapy for young children. For example, emotion regulation and executive functions are tightly linked, and psychotherapy based on play can strengthen executive functions, as is known from diverse fields such as educational kinesthesiology and rehabilitation science [8]. Psychotherapy can impact even somatic functions traditionally considered external to the mind. For example, psychotherapy was found in a small sample to positively impact development from childhood motor deficits [58]. Whereas psychoanalytic psychotherapy once approached medical conditions expansively and retreated, an increasing understanding of its physiological effects and capabilities may again yield promise for applications in promoting its role as a member among a vast array of helpful procedures for children.

Our study therapists were, with one exception, exclusively women. We included measures of the therapists’ countertransference in working with the boys and girls of our trial. Future works may systematically analyze these ratings to observe for measurable differences and utilize analyses of the qualitative data to make observations concerning when gender discrepancies were present or absent, with an eye on implications for technique.

At present, we can conclude that defense-oriented psychoanalytic psychotherapy as represented in RFP-C appears, consistent with prior studies of play psychotherapy, to provide an effective avenue in which to work with boys. No gender differences were found in response rates between boys and girls, despite our finding of a clear effect in ODD symptom reduction as measured by the ODDRS in our randomized controlled trial [36]. Defense-oriented psychoanalytic psychotherapy, organized upon neurodevelopmental principles, which positions boys to make the best use of psychotherapeutic engagement, offers boys an opportunity to grow and to thrive in the safety and security of a long tradition of established therapeutic process.

## Figures and Tables

**Figure 1 behavsci-12-00248-f001:**
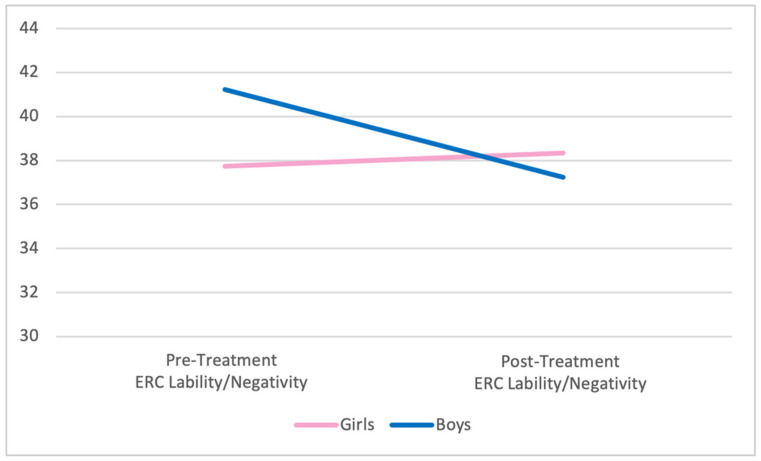
Pre- and post-treatment lability/negativity by gender. Notes. No significant differences comparing pre-/post-treatment or comparing boys to girls. ERC = Emotion Regulation Checklist.

**Figure 2 behavsci-12-00248-f002:**
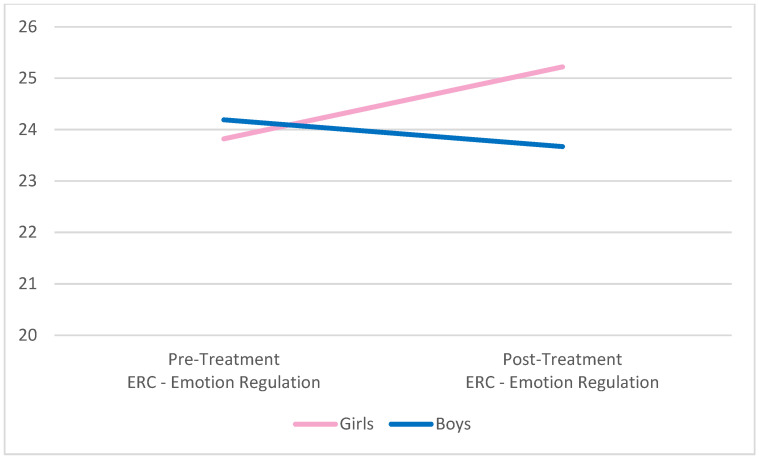
Pre- and post-treatment overall emotion regulation by gender. Notes. No significant differences comparing pre-/post-treatment or comparing boys to girls. ERC = Emotion Regulation Checklist.

**Figure 3 behavsci-12-00248-f003:**
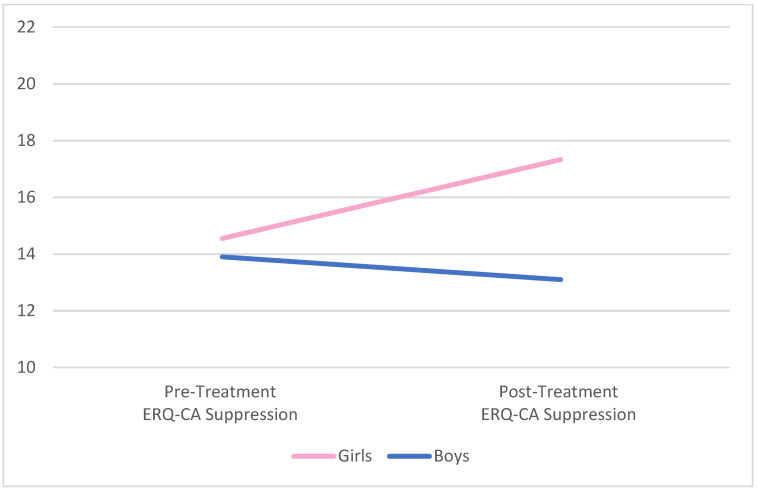
Pre- and post-treatment suppression by gender. Notes. No significant differences comparing pre-/post-treatment or comparing boys to girls. ERQ-CA = Emotion Regulation Questionnaire for Children and Adolescents.

**Figure 4 behavsci-12-00248-f004:**
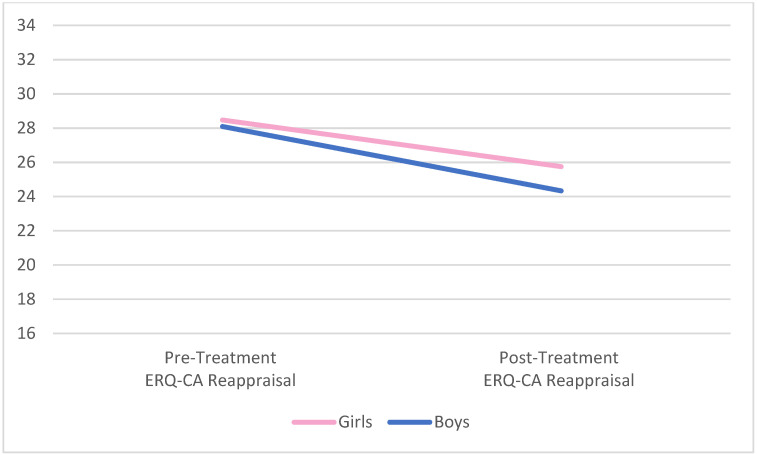
Pre- and post-treatment reappraisal by gender. Notes. No significant differences comparing pre-/post-treatment or comparing boys to girls. ERQ-CA = Emotion Regulation Questionnaire for Children and Adolescents.

**Table 1 behavsci-12-00248-t001:** *Gender Differences at Baseline*.

Independent Samples *t*-Tests and ANOVA				
	Boys (N = 32)	Girls (N = 11)		
	M (SD)	M (SD)	t/F	*p*
Age	7.72 (1.91)	8.18 (2.14)	0.68	0.56
ADHD	64.38 (8.79)	62.83 (7.25)	−0.53	0.60
ODD Symptoms				
CBCL	73.31 (5.30)	70.64 (7.17)	−1.32	0.20
ODDRS	18.59 (3.80)	18.18 (3.25)	−0.32	0.75
ER measures				
ERC neg/lab	40.28 (5.47)	37.18 (3.62)	3.041	0.09
ERC total	24.19	23.82	0.09	0.77
ERQ-CA sup	14.03	13.82	0.01	0.92
ERQ-CA reap	26.90	28.82	0.51	0.48
**Chi-square test**				
	**Boys (N = 31)**	**Girls (N = 11)**		
	**N (%)**	**N (%)**	**χ^2^**	** *p* **
Taking psychiatric medication	7 (23)	0 (0%)	2.98	0.08

Notes. ADHD = attention deficit/hyperactivity disorder; ODD = oppositional defiant disorder; CBCL = Child Behavior Checklist; ODDRS = Oppositional Defiant Disorder Ration Scale; ERC neg/lability = Emotion Regulation Checklist Negativity/Lability subscale score; ERC total = Emotion Regulation Checklist total score; ERQ-CA sup = Emotion Regulation Questionnaire for Children and Adolescents Suppression score; ERQ-CA reap = Emotion Regulation Questionnaire for Children and Adolescents Reappraisal score.

## Data Availability

Data can be obtained upon request.

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
