# Peer review of "Defense-Oriented Psychoanalytic Psychotherapy as a Tailored Treatment for Boys: Neurobiological Underpinnings to Male-Specific Response Tested in Regulation-Focused Psychotherapy for Children"

_behavsci, 2022, doi:10.3390/bs12080248_

Round 1

Reviewer 1 Report

In my opinion, the text is suitable for publication in its present form. The authors have applied all the guidelines mentioned in my previous review.

I wish you all the best!

Author Response

We thank the reviewer for his thoughtful evaluation of the article. A response letter addressing all reviewer suggestions is attached.

Reviewer 2 Report

The presented article on the topic: Defense-Oriented Psychoanalytic Psychotherapy as a Tailored Treatment for Boys: Neurobiological Underpinnings to Male-Specific Response Tested in Regulation Focused Psychotherapy for Children is, in my view, a relatively successful output of a multidisciplinary team of authors. The article is well anchored in the very introduction. Here the authors have a quality base of professional resources. I also appreciate Chapter 1.1 Developmental Hypothesis. It is this part that provides the text with the relevant information needed for a comprehensive view of the issue. I again consider the statistical processing to be successful. The authors present the results. These results are divided into logical areas in a clear follow-up to the previous part of the text. The results are quite remarkable and point to the effect of the intervention procedures that have been chosen.

In the text I miss the final chapter CONCLUSION. It comes to me as a logical connection to the DISCUSSION chapter. Therefore, I ask the authors to complete the conclusion, which will be a summary of the overall work. Ideally, I would also see a suitable quote, for example, thematically focused on comprehensive rehabilitation approaches, where psychotherapy can also be applied. I recommend, for example, the currently published article also in the given magazine (https://www.mdpi.com/2076-3417/12/9/4270). I think that it can be used appropriately and thus extend the conclusion with general information, which relates to individual and comprehensive approaches for children. After all, the world trend is multidisciplinary cooperation, which includes psychology, occupational therapy, special education, etc. I would like such a conclusion with regard to the context of the submitted work.

More specific comments: 

1. The introduction of the presented article can be considered as very well done. As I mentioned. Here the authors choose suitable professional sources and go from general issues to specific ones. There is a clear logical connection and theoretical background for the main part of the presented study.  

2. Statistical processing of results and their presentation is, in my opinion, fine. The title of the article is aimed at male children. The text then shows 32 boys and 11 girls. Why this ratio? Here I miss the point and please explain. Procedures again well processed. I also welcome Chapter 2.4. Thus, the authors chose a descriptive statistician and an ERQ-CA document.  

3. The presentation of the results is at an appropriate level. I will leave it up to the editors to decide whether a subchapter on the results would be appropriate (SUMMARY OF RESULTS), where the authors would summarize in one paragraph otherwise well-processed results. After all, such a summary would be important for readers.  

4. The discussion is fine and contains everything that such a chapter should contain. I have no objection to the adjustment here.  

5. I ask the authors to complete the conclusion. The conclusion should be the generalized outcome of the study. In addition to the above, I also recommend making general conclusions and recommendations for practice. There is no need to break down the conclusion too much. I believe that one or two paragraphs are an appropriate completion of the present study.

Author Response

(The authors gave the same response as above.)

Author Response

We thank the reviewer for his thoughtful evaluation of the article. A response letter addressing all reviewer suggestions is attached.

This manuscript is a resubmission of an earlier submission. The following is a list of the peer review reports and author responses from that submission.

Round 1

Reviewer 1 Report

Dear Authors,

I would like to ask you to describe in more detail the group of researchers - therapists - Why was this group of therapists chosen? How were they selected? In addition, I miss the specification of research hypotheses in points. I would do the same with conclusions.

Congratulations on a very well conducted study.

The paper submitted for review contains important issues from the perspective of behavioral science. The authors have described in detail defense-oriented psychoanalytic psychotherapy as a tailored therapy for boys based on the neurophysiological hypothesis. The manuscript correctly discusses central nervous system development in males, focusing on the development of the emotion regulation system. Through the course of the study, it has been shown that defense-oriented psychoanalytic psychotherapy, structured around neurodevelopmental principles that enable boys to get the most out of psychotherapeutic engagement, provides a unique opportunity for them to grow and develop with the security and confidence that comes from a long tradition of the therapeutic process. The paper deserves praise for its well-organized literature review, choice of research methods, and study group, which is sufficient in works of this nature to make the conclusions presented. In the future, I would suggest conducting similar research on a wider group of people. 

Author Response

Thank you for the positive feedback on our manuscript. A response letter was uploaded.

Reviewer 2 Report

See attached file.

Author Response

Thank you for your suggestions helping to improve the manuscript. A response letter was uploaded.

Round 2

Author Response

A response letter is attached.

Round 3

Reviewer 2 Report

In general, I find the changes to the manuscript not completely satisfying. An additional problematic aspect is that it is not given to know whether this manualized defense-oriented psychoanalytic psychotherapy named Regulation-Focused Psychotherapy for Children had an effect at all, independently from gender. The fact that no difference emerged between genders might depend from the fact that the therapy itself had no effect. Here below are further comments to the manuscript as well as to the authors replies.

·        The authors write: “we additionally edited the text of our discussion section (lines 438-439) to remove our comment upon trends concerning the negativity and lability scores“. However, I cannot find any change in those lines compared to the previous draft of the manuscript, and also not in the earlier lines.

·        The authors introduced the Cohen d at line 158. But how can a so large effect size (> 1)  be not significant? Is this a typo? If not, this should be addressed more clearly.

·        Line 419: the authors wrote “ The post-treatment differences in these scores (p = 0.18) may suggest…”, but 0.18 is not a significant difference and so the text is misleading!

·        I asked in my first reply for the kind of t-test used and thereby meant t-test for dependent vs. independent samples. The authors write “Independent  samples t-tests were used to examine potential differences between group averages. The same approach was used to examine differences in pre- and post-treatment symptom scores for boys and girls.” Does it mean that they used independent sample t-test also for the pre- vs post-treatment comparison?

·        The authors replied that they “specified in greater detail our use of t-tests as two-tailed”; however, I cannot find this statement in the manuscript.

·        Why did the author use a one-way ANOVA although they had 2 factors (gender and treatment-phase)? This should be explained in the Methods. I would expect to find in the results the main effect of Gender, the main effect of Treatment-Phase (pre, post) and the interaction of them, but this is not the case. What was the rational of using a one-way ANOVA instead of a t-test or, even better, of a 2-ways ANOVA?

·        Line 414: I find still hard to understand this sentence: “No differences in emotion regulation differences between the groups pre-treatment”. What does “no differences in…differences”  exactly mean? Please reformulate it.